# Darling: A Web Application for Detecting Disease-Related Biomedical Entity Associations with Literature Mining

**DOI:** 10.3390/biom12040520

**Published:** 2022-03-30

**Authors:** Evangelos Karatzas, Fotis A. Baltoumas, Ioannis Kasionis, Despina Sanoudou, Aristides G. Eliopoulos, Theodosios Theodosiou, Ioannis Iliopoulos, Georgios A. Pavlopoulos

**Affiliations:** 1Institute for Fundamental Biomedical Research, Biomedical Sciences Research Center “Alexander Fleming”, 16672 Vari, Greece; gkasionis2@gmail.com; 2Clinical Genomics and Pharmacogenomics Unit, 4th Department of Internal Medicine, School of Medicine, National and Kapodistrian University of Athens, 11527 Athens, Greece; dsanoudou@bioacademy.gr; 3Center for New Biotechnologies and Precision Medicine, School of Medicine, National and Kapodistrian University of Athens, 11527 Athens, Greece; eliopag@med.uoa.gr; 4Biomedical Research Foundation of the Academy of Athens, 4 Soranou Ephessiou Street, 11527 Athens, Greece; 5Department of Biology, School of Medicine, National and Kapodistrian University of Athens, Mikras Asias 75, 11527 Athens, Greece; 6Department of Basic Sciences, School of Medicine, University of Crete, 71003 Heraklion, Greece; theodosios.theodosiou@gmail.com (T.T.); iliopj@med.uoc.gr (I.I.)

**Keywords:** text-mining, data integration, bioinformatics, named-entity recognition, literature-derived associations

## Abstract

Finding, exploring and filtering frequent sentence-based associations between a disease and a biomedical entity, co-mentioned in disease-related PubMed literature, is a challenge, as the volume of publications increases. Darling is a web application, which utilizes Name Entity Recognition to identify human-related biomedical terms in PubMed articles, mentioned in OMIM, DisGeNET and Human Phenotype Ontology (HPO) disease records, and generates an interactive biomedical entity association network. Nodes in this network represent genes, proteins, chemicals, functions, tissues, diseases, environments and phenotypes. Users can search by identifiers, terms/entities or free text and explore the relevant abstracts in an annotated format.

## 1. Introduction

PubMed^®^ today (02/2022) hosts more than 33 million biomedical abstracts, whereas PubMed Central^®^ Open Access Subset (PMC OA Subset) [1] contains more than 7 Million full-text articles. The ever-increasing amount of literature is posing numerous challenges for bioscientists, as parsing these texts and extracting associations among biomedical entities is neither easy nor trivial. This is particularly true for disease-related research, where a wealth of knowledge on the relations between bioentities (genes, proteins, chemicals, etc.) and pathological conditions is available, especially since the rise of high-throughput experimental methods [2]. There is, therefore, a great need for the development of effective and user-friendly methods for the automated recognition, visualization and analysis of disease-related bioentity associations.

Towards this end, several text-mining approaches have been implemented [3,4,5,6,7]. BioTextQuest [8], for example, retrieves PubMed articles and clusters them based on their biomedical terms. DrugQuest [9] applies text mining on the DrugBank database [10], in order to explore drug associations. DISEASES [11] is a system for extracting disease–gene associations from biomedical abstracts. PREGO [12] uses text mining to link microorganisms with environmental processes and functions. Reflect [13] and EXTRACT [14] perform Named Entity Recognition (NER) on web pages on the fly. FACTA [15] is a text search engine for identifying associated biomedical concepts. OnTheFly [16] parses Office documents, images and PDF files to identify biomedical terms in their text and perform functional enrichment and biological network analysis. CoPub [17] uses Medline abstracts to calculate robust statistics for keyword co-occurrences. NETME [18] offers a knowledge network construction, with term associations in biomedical literature. PubAnnotation [19] is an open, Agile text mining framework to aid researchers throughout the entire annotation process. PubTator [20] provides automated annotations from state-of-the-art text mining systems for genes/proteins, genetic variants, diseases, chemicals, species and cell lines. MetaMap [21] provides access to concepts in the unified medical language system (UMLS) Metathesaurus, from biomedical text. Medline Ranker [22] scores abstracts from Medline, according to a training set of abstracts or a MeSH term. LipiDisease [23] performs disease enrichment analysis on lipids using biomedical literature data. Finally, PESCADOR [24] extracts and analyzes a network of gene and protein interactions from a set of Medline abstracts.

Despite the increasing number of text-mining solutions, effective text mining and analysis of disease-related literature remains challenging. For one thing, the majority of currently available approaches, such as those referenced above, are specialized towards specific bioentity types (e.g., genes, proteins, chemicals, etc.). However, diseases are often complex phenotypes, depending on a multitude of different factors, from gene expression, protein function and chemical substances to cell tissues and even environmental factors. Furthermore, most of these services often offer limited options in the visualization and analysis of their components. To address these challenges, in this article, we present Darling, a novel web application to query scientific publications associated with diseases, identify and visualize bioentities of various types and construct knowledge-based biological interaction networks. Out of a plethora of articles and available databases (reviewed in [25]), we focus on disease-centric repositories and generate a non-redundant set of publications, associated with entries in the OMIM [26], Human Phenotype Ontology (HPO) [27] and DisGeNET [28] databases. The abstracts of the publications are parsed through Named Entity Recognition (NER) to identify a wide range of biomedical terms (genes, chemicals, organisms, ontology terms, diseases, phenotypes and environments). Sentence-based associations among the various biomedical entities are presented in an interactive network [29,30], as well as in searchable and sortable tables, while abstracts are shown in annotated format. Statistics regarding the frequencies of the queried entity types are also presented. Darling is available at http://darling.pavlopouloslab.info or http://bib.fleming.gr:8084/app/darling (accessed on 28 February 2022).

## 2. Materials and Methods

### 2.1. Data Collection

The database records (October 2021 data) of OMIM (25,767 entries), HPO (4645 entries) and the human subset of DisGeNET v. 7.0 (30,170 entries) were parsed and their associated publications were isolated, resulting in a non-redundant set of 881,185 articles. The article abstracts were retrieved from PubMed using the Entrez Direct API [31] and were analyzed through NER to isolate bioentities, using the EXTRACT tagger [14,32]. The EXTRACT tagger uses a dictionary-based approach, through which biological and biomedical terms, both canonical and synonyms (e.g., gene name aliases), are assigned to their unique identifiers; thus producing concept-normalized results. The extracted bioentities were assigned to their proper database identifiers, resulting in a non-redundant set of 78,938 terms including genes (protein-coding and other gene types, e.g., micro-RNAs), chemical compounds, Gene Ontology terms, tissues, diseases, organisms, phenotypes and environments. In the dataset, each term is represented by its unique identifier to the relevant database (Table 1), its canonical name, and a number of alternative names/synonyms, as found through the mining of the publications. A knowledge-based interaction network was constructed from these terms (nodes), using their co-occurrence to define interactions (edges). Specifically, two terms were defined as interaction partners if they were mentioned in the same sentence in the text, with their edge weight defined as the sum of the two terms’ co-mentions in the analyzed abstracts. The aforementioned approach resulted in knowledge network consisting of 78,938 nodes and 5,235,076 edges. Table 1 summarizes the number of biomedical terms identified for each category. A flowchart demonstrating the data retrieval and analysis procedure is shown in Figure 1.

### 2.2. Darling Application and Analysis

***Query:*** Darling’s GUI, offers different query options through three tabs. These are: *(i) Disease Search, (ii) Bioentity Search* and *(iii) Literature Search.*


In the first case, one can directly query any of the OMIM, HPO and DisGeNET databases using their original identifiers or disease names. Users can only query one of the databases each time but, with each query, one can append the article result list for further analysis. Duplicated retrieved terms are discarded in the next step of the analysis. In the case of free text querying (e.g., disease name), users can force Darling to look for exact matches or substrings in the database’s record names. In the case of using database identifiers, users can use lists of IDs separated by spaces or commas to retrieve the results of multiple disease entries.

In the second tab, one can search for a bioentity term using free text and exact or partial string matches. In this case, the user can search for chemicals, proteins or tissues stored in Darling’s database and perform a non-disease-centric analysis from a different starting point (e.g., a chemical). Notably, exact matches refer to the bioentity terms identified by the EXTRACT tagging service.

The third option is the most flexible as one can use a list of PubMed identifiers or free text to look for exact or partial matches in article titles. In this case, Darling will search for terms (e.g., “CRISPR-Cas9” or “mir-19”) that may not appear in its dictionary or any of the OMIM, HPO and DisGeNET record names.

In every case, after submitting a search query, Darling will fetch all matched articles (Figure 2). The collected articles are then summarized in an interactive and sortable table for review prior to further analysis; thus, users may either keep all retrieved articles or focus on a subset. When multiple search queries are executed, the results of each query can also be filtered to include the intersection (only the common results) or union (all results) of the queries. Users may also choose to filter the NER results and subsequently the retrieved associations by selecting one or more bioentity types (genes, proteins, chemicals, functions, tissues, diseases, environments and phenotypes). Notably, all of these actions can be applied on the set of 881,185 articles mentioned in OMIM, HPO and DisGeNET databases. OMIM’s body text is not processed due to license restrictions.

***Tables and statistics:*** Upon selecting articles and applying entity-type filters, Darling will mine all articles of interest and retrieve the corresponding NER results from its database (pre-calculated with the use of EXTRACT [14]). Identified terms are reported in searchable and sortable tables along with their synonyms, official symbols, database identifiers and links to the original source. Identified terms can be reported altogether or separately in corresponding tabs—one per category (genes/proteins, chemicals, functions, tissues, diseases, environments and phenotypes). Extra columns indicate how many times a term was found in the retrieved abstracts as well as in how many articles this term was detected. Interactive ordered bar blots are generated to show such frequencies while interactive pie charts show the overall coverage of terms and articles retrieved for every bioentity category. Finally, word clouds show the most common terms, scaled by their frequency.

***Network:*** In addition to the tables, Darling generates an interactive association network of the identified bioentities. Network nodes may fall into any of the identified bioentity types (genes/proteins, chemicals, organisms, GO terms, tissues, diseases, environments and phenotypes) and are assigned a certain color (distinct per category). Node sizes can be adjusted according to how many times they were identified in a selected set of abstracts (total frequency) whereas network edges can be interactively filtered according to the total times two adjacent entities were located in the same sentence (edge weight). At any stage of analysis, users may limit the visible nodes to certain bioentity types. For aesthetical convenience, users may adjust the network view using various offered layouts [45]. Characteristic examples are the force-directed ones such as Fruchterman–Reingold [46] and Kamada–Kawai [47] or the plain ones such as grid, random and circular layout. The network is fully interactive and comes with control buttons for positioning, zooming and recentering. Nodes can be dragged and positioned anywhere on the plane. 

Besides visualization, similarly to the NAP application [48,49] or Cytoscape’s Network analyzer [50], Darling offers basic network topological analysis where users can see numerical values for the numbers of nodes and edges, density, modularity, radius, average path length, average connectivity, average clustering coefficient, betweenness and eccentricity centrality. 

For more comprehensive visualization and analysis, at any stage, the network’s edge list can be exported in a tab-delimited file format and visualized with external viewers [51,52,53] (e.g., Cytoscape [54], Gephi [55], NORMA [56], Arena3D^web^ [57]). Bidirectional edges (e.g., AB-BA) are kept only once. 

***Annotated text:*** At any stage of the analysis, the relevant PubMed article abstracts are reported in an annotated format in a separate table. Users can read these abstracts with the identified terms highlighted in different colors according to the tagged entity category. On mouse-hovering or clicking over a term, a popup window with relevant links to the corresponding databases is generated on-the-fly.

***Functional Enrichment:*** After the network generation and the application of any filtering options, all of the visible identified genes and proteins can be sent to the Flame application [58] for functional and literature enrichment analysis. Genes and Proteins will be first converted to ENSEMBL identifiers and can then be analyzed for KEGG [59,60], Reactome [61,62] and Wiki Pathways [63] or for the biological functions [37] they are involved in. Flame utilizes g:Profiler [64] and aGOTool [65] at its backend for functional and literature enrichment and offers appealing visualizations for easier interpretation of the reported results. In addition, Flame can construct protein–protein interaction networks, by retrieving evidence from the STRING database [66].

### 2.3. Implementation

Darling is organized in a *MySQL* database which is periodically updated. The GUI and backend are mainly written in *R/Shiny*. The interactive network is visualized with the *R/visNetwork* library and network topological analysis is performed using the *R/igraph* library [67]. Plots are generated with the use of *R/Plotly* [68], while wordclouds with the *R/wordcloud2* library. The *EXTRACT* API [14] is utilized to display popup windows for bioentity terms in the annotated abstracts.

## 3. Results

### 3.1. Investigating the Link between Obesity and Cardiovascular Diseases

To demonstrate Darling’s capacity for the extraction of biological information and knowledge discovery, we investigated genes and pathways that may link cardiovascular disease (CVD) to obesity. We queried DisGeNET, using the disease term “cardiovascular”, and obtained the 5000 most recent articles, 100 of which also contained the term “obesity”. This group entailed 317 entities (Figure 3A) that included 109 unique genes/proteins associated with “insulin receptor signaling”, “metabolic disease”, “energy homeostasis” and “cytolysis” gene ontology (GO) biological processes (Figure 3B). By using Darling, we constructed a co-occurrence network of genes, phenotypes and tissues predicted to link CVD to obesity (Figure 3C). A major neighborhood in this network (subnetwork 1) is associated with “insulin resistance”, “abnormal inflammatory response” and “cardiac hypertrophy” and linked to the adipose tissue, liver and blood. This group mostly entails components of the adiponectin pathway, including adiponectin (ADIPOQ), its receptors ADIPOR1 and ADIPOR2, and their downstream adaptors APPL1/2, which transduce the anti-atherogenic and anti-inflammatory effects of adiponectin. The group also includes the inflammation marker CRP, which is elevated in both obesity and CVD, and GAS6, which has been implicated in atherosclerosis, thrombosis and innate immune reactions [69]. 

The Darling co-occurrence network also indicated an interaction between fat mass and obesity-associated protein (FTO) and apolipoprotein E (APOE), linked to inflammatory processes (Figure 3C). Several FTO polymorphisms are associated with increased risk for weight gain [70] and the APOE ε4 variant is a genetic risk factor for atherosclerosis and CVD in humans [71]. Experimental evidence suggests that expression of APOE ε4 leads to elevated intracellular and circulating cholesterol levels and heightened inflammatory reactions compared to other variants [71]. A putative mechanistic link between APOE and FTO is underscored by studies showing that overexpression of FTO in APOE-deficient mice reduces cholesterol and inflammatory cytokine synthesis by macrophages and alleviates atherosclerosis associated with the absence of APOE [72].

Another neighborhood of interest identified by Darling (subnetwork 2; Figure 3C) entails the growth differentiation factor 15 (GDF15) and its receptor, GFRAL. Circulating GDF15 crosses the blood brain axis to bind GFRAL in neurons of the hindbrain, leading to reduced appetite and food intake. The serum levels of GDF15 dramatically increase in cancer-associated cachexia but are also found elevated in obesity, presumably acting as a compensatory mechanism to reduce appetite [73]. GDF15 has been reported as a prospective biomarker of CVD and independent predictor of all-cause mortality [74,75]. Interestingly, CRP has been found to induce the expression of GDF15 [76]. 

Overall, the aforementioned observations demonstrate the capacity of Darling for the extraction of biological information and knowledge discovery.

### 3.2. Querying Multiple Disease Databases Simultaneously with Darling

In a second case study, we asked whether Darling could facilitate the extraction of biological information on a disease by combining several disease libraries. To this end, we interrogated OMIM, HPO and DisGeNET for Cornelia de Lange, a rare genetic syndrome characterized by slow growth rates, leading to short stature, intellectual disability that ranges from moderate to severe, congenital heart defects and bone abnormalities, among others. Through Darling, we queried OMIM for “Cornelia de Lange” and obtained 127 entries that generated 264 entities. By exploring the same query against the OMIM, HPO and DisGeNET compendiums together, Darling retrieved 318 entries that generated 292 unique articles and 712 entities. The co-occurrence network of genes/proteins, GO Biological Processes and DOID diseases derived by these 712 entities yielded superior information compared to the respective network derived from OMIM only (Figure 4).

A major neighborhood in both networks (Figure 4A,B) contained the NPBL, SMC1A, SMC3, HDAC8 and RAD21 genes, which are found mutated in >80% of Cornelia de Lange patients. These genes encode for regulators of the cohesin complex and are involved in chromosome condensation, chromosome segregation and DNA repair. Additional genes found exclusively in the network generated from OMIM, HPO and DisGeNET include BRD4 and MAU2. Mutations in both genes have recently been detected in Cornelia de Lange patients and have been functionally implicated in disease pathogenesis through their interaction with NPBL [77,78].

Interactions with Wilson–Turner and Roberts syndromes were also indicated in this network (Figure 4B). Roberts syndrome bears developmental abnormalities, similar to Cornelia de Lange, such as limb abnormalities, retarded growth and intellectual impairment. Mechanistically, Roberts syndrome has been linked to mutations in ESCO2 gene, which encodes a cohesin acetyltransferase and modulator of double strand break repair [79]. Wilson–Turner syndrome is a rare X-linked multisystem genetic disease that also manifests with intellectual disability, dysmorphic facial features and short stature and has been linked to a mutation in the HDAC8 gene [80]. Overall, the aforementioned examples demonstrate the capacity of Darling for the extraction of biological information and acceleration of knowledge discovery.

## 4. Discussion

Darling is a text-mining application, aiming to aid researchers in associating different biomedical entities in a knowledge network, generated by literature mining. A great advantage of Darling is its high quality back-end NER tagger, which makes it more competitive compared to other similar applications, both in terms of annotation and data integration. In addition, Darling only focuses on a subset of disease-centric articles, which have been manually curated in the OMIM, HPO and DisGeNET databases, rather than the whole PubMed space. Taking into account that PubMed currently contains many review articles and has also recently started to support preprints [81], we believe that this is the safest approach, in order to eliminate possible false-positive term associations. Nevertheless, we plan to extend Darling’s functionality in the future and cover literature coming from more databases, as well as support full text articles.

In its core, Darling contains a relational database, consisting of all relevant bioentity information and associations. Term frequencies per article, their respective canonical names and the relative tagged documents are all pre-calculated, further speeding up the execution of the application, and are served via an interactive GUI. Therefore, Darling does not depend on external web services, as opposed to other similar applications (e.g., NETME), which query the various databases (e.g., PubMed) on the fly, resulting in time-consuming requests. This may secure an always up-to-date information schema but comes at the cost of speed, performance and web-service dependencies. To keep up to date, Darling’s database will be annually updated, including new OMIM, HPO and DisGeNET entries, as well as their associated publications and extracted bioentities. Furthermore, in future versions, Darling will implement additional databases, support more model organisms and enable the detection of abstract-based associations (currently only offers sentenced-based), something which may increase the network’s complexity.

Overall, we believe that Darling outperforms most of the currently available tools, in terms of performance, variety of identified entity terms and quality of results. It is a powerful tool, which can simplify the way researchers query and explore existing knowledge, while also identifying novel indirect associations among biomedical entities, which may be the pivot elements for new hypotheses and discoveries.

## Figures and Tables

**Figure 1 biomolecules-12-00520-f001:**
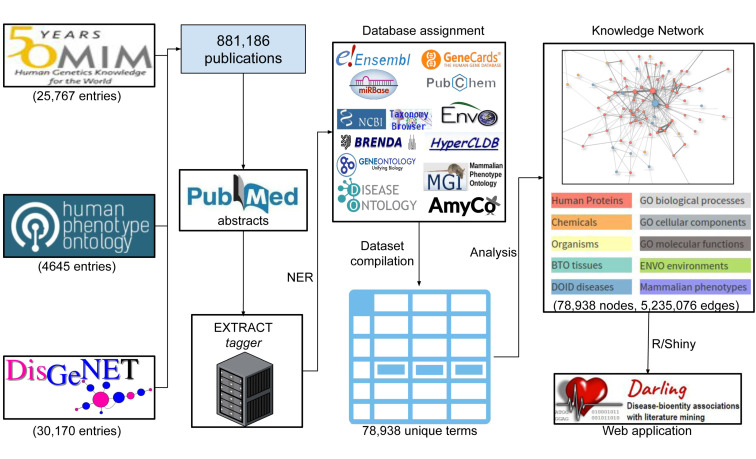
Flowchart of the data retrieval procedure implemented in Darling.

**Figure 2 biomolecules-12-00520-f002:**
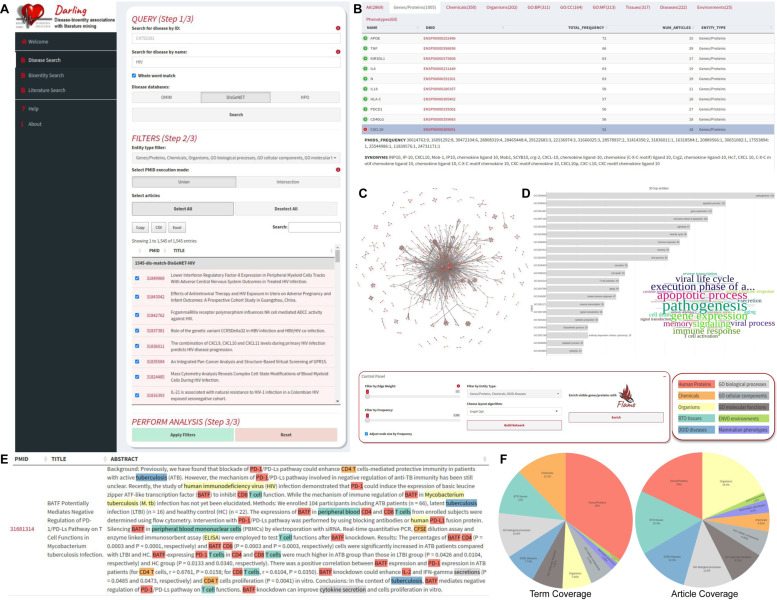
The Darling Graphical User Interface of Darling. (**A**) The input form of the *Disease Search* query. Users can perform searches using a disease’s name or database identifier, against the data retrieved from OMIM, HPO or DisGeNET. In the example, the term “HIV” is searched against DisGeNET. The search form initially returns the publications associated with the disease. Users can then choose the publications and entity types of their interest and perform an analysis, using the form elements at the bottom of the page. (**B**) Excerpts of the results retrieved for the search. A total of 2869 entities have been retrieved and organized in distinct categories. For each term, its database identifier, canonical name, synonym terms and associated PubMed identifiers (PMIDs) are shown; in addition, the frequency of the terms, both total and for each distinct publication, is calculated. (**C**) A knowledge-based association network generated by the search results. Users can adjust the elements of the network and select from a list of different visualization layouts. (**D**) Bar-plot (left) and word cloud (right) representations of the most frequent GO biological processes associated with HIV. (**E**) A graphical representation of one of the publication abstracts, with extracted bioentities highlighted in color. (**F**) Pie charts of the overall coverage of terms and articles retrieved for every bioentity category.

**Figure 3 biomolecules-12-00520-f003:**
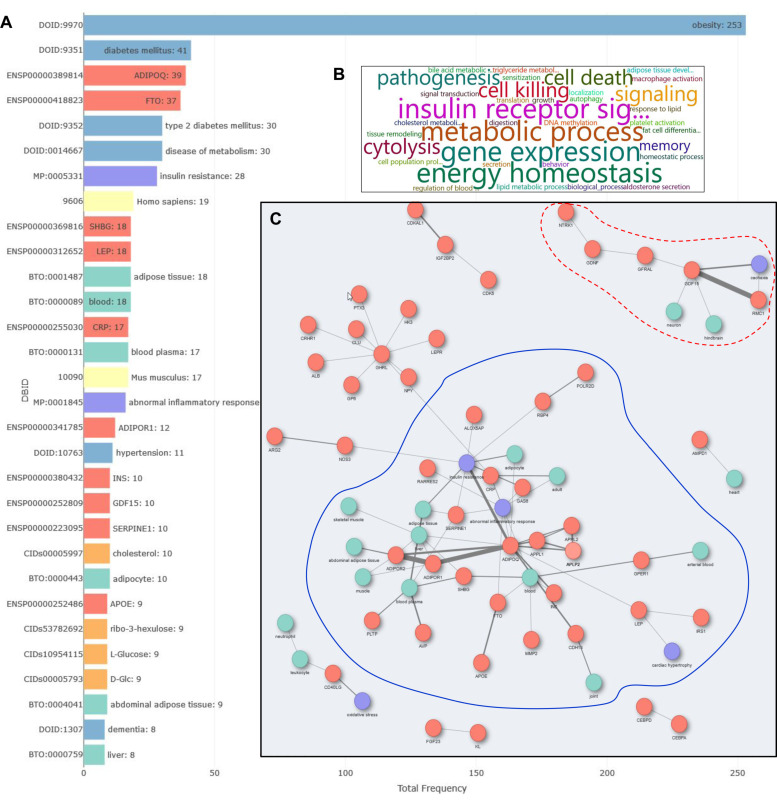
Assessment of Darling’s capacity for extraction of biological information and knowledge discovery. (**A**) Top entities (chemicals, phenotypes, proteins/genes, organisms, GO:BP, GO:CC and GO:MF, tissues, environments and diseases) plotted against frequency of occurrence in 100 DisGeNET-related publications. (**B**) GO Biological processes predicted to be mostly associated with cardiovascular disease. (**C**) Co-occurrence network depicting genes (orange circles), phenotypes (purple circles) and tissues (green circles) predicted to link cardiovascular disease and obesity. Subnetwork 1, discussed in the Results section, is demarcated by the blue line and subnetwork 2 by the red line.

**Figure 4 biomolecules-12-00520-f004:**
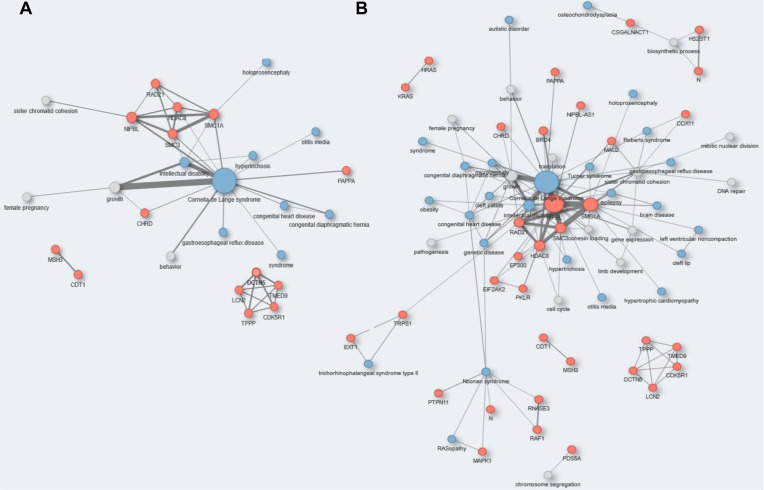
Networks of genes/proteins, GO Biological Processes and DOID diseases generated by Darling query of OMIM (**A**) or OMIM, HPO and DisGeNET compendiums together (**B**) for “Cornelia de Lange”. Entities (264 and 712, respectively) were used to build co-occurrence networks (filter by frequency = 5).

**Table 1 biomolecules-12-00520-t001:** Identified biomedical terms in a set of 881,185 articles mentioned in OMIM, HPO and DisGeNET databases.

Entity Type	Resource	#Terms
Chemicals	PubChem [33]	23,593
Genes/Proteins	ENSEMBL [34], miRBase [35], Gene Cards [36]	19,731
GO—Biological Process	Gene Ontology [37]	6002
GO—Molecular Function	Gene Ontology [37]	3176
GO—Cellular Component	Gene Ontology [37]	1842
Tissues	BRENDA Tissue Ontology (BTO) [38]	4229
Diseases	Disease Ontology [39], AmyCo [40]	6172
Organisms	NCBI Taxonomy [41]	11,212
Environments	Environmental Ontology (ENVO) [42]	363
Phenotypes	Mammalian Phenotype Ontology [43], Cell Line Data Base (CLDB) [44]	2618

## Data Availability

Darling is available online at http://darling.pavlopouloslab.info (accessed on 28 February 2022).

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
