# Peer review of "Darling: A Web Application for Detecting Disease-Related Biomedical Entity Associations with Literature Mining"

_biomolecules, 2022, doi:10.3390/biom12040520_

Round 1

Reviewer 1 Report

This article introduced a new web application for detecting disease-related biomedical entity associations with literature mining. The web application is public available and user-friendly. There are several questions,  which need clarification from the authors.

  1. The authors collected date from a few database. While the database have multiple versions, was the latest version used in this web application? Could the web application list all database name and version/date somewhere in the website, as well as its own updates/releases information? 
  2. For gene search, how were the gene symbol aliases handled? Publications may use different gene aliases for a gene. Could the web application get all result?
  3. Is there a function can search a list of diseases or bioentities?
  4. Is there a legend available in the web for the generated network and other plots?
  5. I searched "FTO" using Bioentity search. But returned nothing. As mentioned in the article, FTO is a known obesity-associated protein. Why was there no search result?

Author Response

Reviewer 1

This article introduced a new web application for detecting disease-related biomedical entity associations with literature mining. The web application is public available and user-friendly. There are several questions,  which need clarification from the authors.

Response: We thank the reviewer for their comments and criticism, which have helped improve the manuscript as well as the Darling web application.

Point 1. The authors collected date from a few database. While the database have multiple versions, was the latest version used in this web application? Could the web application list all database name and version/date somewhere in the website, as well as its own updates/releases information? 

Response: We thank the reviewer for this comment. In the paper, it is stated that all data are based on the October 2021 versions of the OMIM, HPO and DisGeNET databases (Section 2.1, “Data Collection”). With regards to updates, we plan to conduct annual updates to Darling’s underlying data, including database entries, PubMed publications and all associated text mining results.

A sentence clarifying the update plan has been added to the manuscript (Section 4, “Discussion”).  In addition, the Darling application now also includes a statistics overview in its home page, displaying the contents from each database and also showing their versions (date of data retrieval).

Point 2. For gene search, how were the gene symbol aliases handled? Publications may use different gene aliases for a gene. Could the web application get all results?

All entity terms, including aliases/synonyms, are handled by EXTRACT’s text mining tagger in a dictionary-based approach.  In this approach all text-mined terms, both canonical and synonyms, are assigned to database and ontology identifiers; thus producing concept-normalized results. This essentially means that all synonyms/aliases of a term are assigned to the same unique bioentity, so that there are no erroneous multiple assignments (i.e. aliases of the same entity treated as different bioentities).

A sentence clarifying this has been added to the revised manuscript (Section 2.1, “Data Collection”).

For example, the aliases “EGFR”, “Erbb1”, “Her1”, “epidermal growth factor receptor” etc will all correspond to the same bioentity (the EGFR gene/protein), represented by a unique ID (ENSP00000275493).  In Darling’s results, the entity is represented by its canonical name (EGFR), its identifier (ENSP00000275493) as well as its synonyms as they have appeared in the analyzed literature (available in the “synonyms” column of the results, which can be viewed by clicking on the “+” icon on each table row). More importantly, queries are performed on all term aliases, meaning that users searching for EGFR can use its various synonyms (“EGFR”, “Her1”, “Erbb1” etc) and obtain results.

Point 3. Is there a function can search a list of diseases or bioentities?

Response: Users can search the results for lists of diseases or publications simply by providing a list of the relevant identifiers (e.g. DisGeNET disease IDs or PubMed PMIDS) in the search, separated by commas or spaces. The results of all searched terms are returned (provided of course they exist in Darling’s database). This function is currently offered for the Disease and Literature searches, with disease identifiers (OMIM, HPO, DisGeNET) or PubMed PMIDs, respectively.

A sentence clarifying this has been added to the revised manuscript (section 2.2, “Darling application and analysis”).

Future versions of Darling will also support searching for multiple bioentities in a similar manner.

Point 4. Is there a legend available in the web for the generated network and other plots?

Response: A color legend for the network appears in the panel titled “color-coding”.  The same color scheme is also used for the annotated documents tab, as well as all plots (bar plots and pie charts). Color legends appear in the Network, Annotated Documents and Query statistics tabs; in addition, the color legend has been added above the results in the Identified Terms tab, accessible by clicking the Expand/Collapse (“+”) button.

Point 5. I searched "FTO" using Bioentity search. But returned nothing. As mentioned in the article, FTO is a known obesity-associated protein. Why was there no search result?

Response: We thank the reviewer for raising this point. In all search categories, including Bioentity, Darling performs searches with an option that can be activated or deactivated, called “whole word match”. When this option is activated, the input term  (e.g. “FTO”) is considered a full, distinct word, and only results with an exact match are returned. When it is deactivated, the input term is also searched as a subset (e.g. “FTO-related”, “FTO-specific” etc.) and all relevant results are returned.  This was also the case with the reviewer’s query, for which searches without “whole word match” return results, while queries with the option enabled don’t.

In the original Darling version, this option was activated by default in all queries, and users would have to disable it by themselves to perform more general queries.  However, after taking this point into account, we have decided to leave “whole word match” deactivated by default, so that users can select to perform distinct word matches only when they need to.

Reviewer 2 Report

Dear Editors,

Dear Authors,

The reviewed manuscript entitled: “Darling: A web application for detecting disease-related biomedical entity associations with literature mining” represents very valuable study. The manuscript introduces new bioinformatics tool that offers the assistance in data mining for researchers aimed to human disease investigation. The authors have proved in the reviewed manuscript the power of developed tool in detection of association between disease and biomedical entities. In my opinion the software is very helpful and has great potential in future applications as the amount of data increases fast, making difficult its efficient analysis. I have checked all described functionality of the application and results are always reliable and replicable. Application is very intuitive in work and easy result interpretation. During testing I have found some functional problem with the text annotation (section “Annotated Documents”), where after clicking over highlighted term, a popup window does not provide the information. Maybe it is problem with my OS platform – I am not sure. Regarding to manuscript, it is written very well and in understandable way – I do not have any remarks in this regard. I hope the authors will be systematically updating the database, develop new functionality, i.e., abstract-associated analysis and manuscript full text support. Maybe one day the application will offer data mining not only for human disease but also for animal genetics and disease, including fish.

In conclusion, I highly recommend to publish the reviewed manuscript in the Biomolecules Periodical.

My congratulations for the Authors, very good job!

Thank you for another interesting manuscript that I could review!

Author Response

Dear Authors,

The reviewed manuscript entitled: “Darling: A web application for detecting disease-related biomedical entity associations with literature mining” represents very valuable study. The manuscript introduces new bioinformatics tool that offers the assistance in data mining for researchers aimed to human disease investigation. The authors have proved in the reviewed manuscript the power of developed tool in detection of association between disease and biomedical entities. In my opinion the software is very helpful and has great potential in future applications as the amount of data increases fast, making difficult its efficient analysis. I have checked all described functionality of the application and results are always reliable and replicable. Application is very intuitive in work and easy result interpretation. During testing I have found some functional problem with the text annotation (section “Annotated Documents”), where after clicking over highlighted term, a popup window does not provide the information. Maybe it is problem with my OS platform – I am not sure. Regarding to manuscript, it is written very well and in understandable way – I do not have any remarks in this regard. I hope the authors will be systematically updating the database, develop new functionality, i.e., abstract-associated analysis and manuscript full text support. Maybe one day the application will offer data mining not only for human disease but also for animal genetics and disease, including fish.

In conclusion, I highly recommend to publish the reviewed manuscript in the Biomolecules Periodical.

My congratulations for the Authors, very good job!

Thank you for another interesting manuscript that I could review!

Response: We thank the reviewer for his/her positive comments and review. With regards to the annotated documents tab, we would like to state that we have tested the feature in all major browsers (Chrome, Edge, Firefox etc) without any issues.  However, we should point out that this particular information is presented through pop-up windows which may be blocked by pop-up blocking add-ons, extensions or other similar browser settings.

Reviewer 3 Report

An interesting and sound manuscript describing a new web application for searching biological entities related to a disease.
The webapplication is easy to use, finds effectively the requested information, and has a pleasant user interface.
The manuscript is well written and structured.
My only suggestion is to add another example of use in section results.

Author Response

An interesting and sound manuscript describing a new web application for searching biological entities related to a disease.

The web application is easy to use, finds effectively the requested information, and has a pleasant user interface.

The manuscript is well written and structured.

My only suggestion is to add another example of use in section results.

Response: We thank the reviewer for their comment. As per their suggestion, a second case study has been added in the revised version of the manuscript (see section 3.2, “Querying multiple disease databases simultaneously with Darling”).

Round 2

Reviewer 1 Report

Thanks for addressing the previous comments.